# Algorithm, expert, or both? Evaluating the role of feature selection methods on user preferences and reliance

**Jaroslaw Kornowicz** [ID]*, **Kirsten Thommes** [ID]

Faculty of Business Administration and Economics, Paderborn University, Paderborn, Germany

* jaroslaw.kornowicz@uni-paderborn.de

## Abstract

The integration of users and experts in machine learning is a widely studied topic in artificial intelligence literature. Similarly, human-computer interaction research extensively explores the factors that influence the acceptance of AI as a decision support system. In this experimental study, we investigate users' preferences regarding the integration of experts in the development of such systems and how this affects their reliance on these systems. Specifically, we focus on the process of feature selection—an element that is gaining importance due to the growing demand for transparency in machine learning models. We differentiate between three feature selection methods: algorithm-based, expert-based, and a combined approach. In the first treatment, we analyze users' preferences for these methods. In the second treatment, we randomly assign users to one of the three methods and analyze whether the method affects advice reliance. Users prefer the combined method, followed by the expert-based and algorithm-based methods. However, the users in the second treatment rely equally on all methods. Thus, we find a remarkable difference between stated preferences and actual usage, revealing a significant attitude-behavior-gap. Moreover, allowing the users to choose their preferred method had no effect, and the preferences and the extent of reliance were domain-specific. The findings underscore the importance of understanding cognitive processes in AI-supported decisions and the need for behavioral experiments in human-AI interactions.

## Introduction

As artificial intelligence (AI) becomes increasingly powerful through advances in computing power, improved algorithms, and the availability of more data, its prevalence expands across a wide array of fields and life situations [1–5]. In response to this growing ubiquity, recent research efforts have shifted from solely focusing on improving the accuracy of AI models to addressing the interaction with a more diverse and heterogeneous user base, exploring the potential consequences of AI adoption and understanding users' preferences and concerns [6].

**Data availability statement:** Experimental data and analysis scripts can be found at https://osf.io/z2xpy/?view_only=90607651bed949d29593c4a176d6c96d.

**Funding:** We gratefully acknowledge funding by the German Research Foundation (Deutsche Forschungsgemeinschaft, DFG): TRR 318/1 2021 – 438445824.

**Competing interests:** NO authors have competing interests.

One strand of research focuses on the human user and has observed that user reliance on algorithmic decision aids is not uniform and is influenced by various factors [7,8] such as the user's personality, algorithm design, task factors, and high-level factors as organizational and societal aspects. The literature surrounding "algorithm aversion" has documented a stated preference among users for human decision-making over algorithmic advice and has noted that individual aspects of AI systems can impact trustworthiness and reliance [7–10]. However, these results encounter resistance, often described as "algorithm appreciation" that observes the converse—a stated preference in favor of algorithms [11,12].

Another stream of research has concentrated on the system, enhancing transparency and explainability as methods to make AI more accessible, comprehensible, and reliable [13]. Legal institutions also drive this research landscape. The increasing presence of AI in society has prompted governments to establish requirements for greater transparency [14,15]. These regulations have led to "black box" models becoming more informative to end users, with implications for AI reliance among all stakeholders. In addition, interdisciplinary efforts between computer scientists, social scientists, and ethicists are increasingly encouraged to tackle the complex challenges posed by AI integration in society [16,17].

Instead of explaining the model or the outcome, recent research discusses other means of quality control during the development of the AI system, e.g., adding human agency. The basic idea here is that not every user must be able to understand the system, but that experts, e.g., domain experts, are involved in the process of machine learning (ML) development, supervise the system, and add human expert knowledge—resulting in a more trustworthy ML models for every end user [18–20].

Previous research has highlighted the significance of human involvement and its effect on users' perceptions, preferences, and reliance. It can be categorized in two ways: involvement in the development and training (typically beyond the scope of the user) and the degree to which humans can apply AI, giving the user options on how to utilize recommendations for their decisions [7]. Limited research has been directed towards the former. Ashoori and Weisz [9] and Jago [21] demonstrated that users tend to favor models trained by data scientists or experts instead of those trained autonomously, without explicitly specifying the nature of the involvement. In a recent study that inspired our work, Cheng and Chouldechova [22] involved users at various stages. They discovered that permitting users to select the training algorithm can mitigate aversion, whereas modifying the inputs does not. While a detailed description of human involvement may not be necessary in many cases, it can be essential in highly transparent models, where features are readily visible, such as in scoring systems [23]. The literature review by Jussupow et al. [7] reveals that it is important to note that human responses differ between the stated preferences and the chosen behavioral response, i.e. their actual reliance. While many studies find a strong preference for human oversight, the revealed preferences in terms of actual behavior as less clear. In our study, we set out to analyze whether stated and revealed preferences are aligned.

Although there are many areas for human involvement, in this paper we focus on the role of human involvement within feature selection. Feature selection is a pivotal step in the machine learning pipeline. It involves identifying the most relevant variables from the input data, which can significantly impact the predictive performance and interpretability of the resulting model [24,25]. Algorithmic feature selection methods are often criticized for lacking theoretical or expert knowledge. Consequently, many scholars argue for human-based feature selection methods or a collaboration of algorithms and humans for feature selection and other machine learning processes [25–27]. We contribute to answering this call.

In our study, we distinguish three methods of feature selection: algorithm-based feature selection (*Algorithm*), expert-based feature selection (*Expert*), and a combined approach (*Combination*). We seek to answer three research questions:

1) What kind of feature selection method do users prefer?
2) Does the feature selection method affect reliance?
3) Does allowing the user to choose their preferred method affect reliance?

Yet, as far as we know, the question of how feature selection modes contribute to AI reliance has not been systematically analyzed. Nonetheless, feature selection and human preferences for feature selection mechanisms are crucial to understanding a model. The novelty of our study lies in addressing the gap in the literature by examining the effects of different levels of human integration in feature selection on user preferences and reliance.

To answer our questions, we conducted an online study involving 216 participants. Our results reveal that *Combination* was the most preferred, followed by *Expert* and *Algorithm*. However, these relationships vary depending on the task domain. Interestingly, stated preferences do not correlate with behavioral reliance, similar to previous studies [28,29]. In a second treatment, we randomly allocate a new group of users to models whose features are either selected by *Expert*, *Algorithm*, or a *Combination*. We observe no significant effect of the underlying feature selection methods on advice reliance. Moreover, the involvement of participants in choosing their preferred feature selection method does not affect the reliance. Reliance is also different across domains. We find a significantly higher probability of reliance in the medical domain compared to a sports-related domain. Concerning individual differences, we observe that participants displaying higher risk-taking tendencies prefer *Algorithm* and *Combination* over *Expert*.

Our study underscores the value of behavioral experiments with incentivized tasks in understanding human-AI collaboration. It points to the importance of further examining cognitive processes in decision-making with AI assistance and stresses the challenge and importance of considering domain-specific effects.

## Related work

### Feature selection

A critical process in developing ML models is feature selection [30]. Features, also called predictors, variables, dimensions, or inputs, can be defined as measurable properties or characteristics of observed procedures or entities [31,32]. Selecting an appropriate subset of features for an ML model can significantly impact its performance, interpretability, computation time, and overfitting risk [33]. This is especially relevant for high-dimensional datasets, which may contain irrelevant and redundant features that negatively affect the quality of the learned models for stakeholders [34]. Feature selection can be used for simple tabular datasets, but also for image data, for example, to improve super-resolution algorithms [35] or computer-aided diagnosis for glaucoma identification [36] and cancer prediction [37].

The domain of feature selection is extensively studied, with the development of various automated algorithms that aim to select relevant feature subsets from datasets [38]. Feature selection techniques driven by data can be generally divided into three categories: filter methods that assess features solely based on the data; wrapper methods that select features through the predictive capability of a machine learning algorithm; and embedded approaches such as LASSO regression that come with inherent feature selection processes [24]. There are also

hybrid methods that show great promise, indicating that research in this area continues to grow [39].

Equally relevant to our research is incorporating human knowledge in feature selection, sourced directly from domain specialists or literature. For instance, Naher et al. [40] demonstrated that features based on a literature review significantly improved the accuracy of a heart disease classifier. Human knowledge-driven feature selection can involve researching relevant scholarly literature [40–42] or consulting domain experts [43,44]. These approaches are particularly important for model explainability, ensuring that the selected features do not contradict human knowledge [45].

It is also feasible to combine various approaches. Multiple feature sets, potentially sourced from different origins, can be aggregated into a singular final set [46,47]. Additionally, there are interactive methodologies wherein humans and algorithms collaborate iterative [48,49]. Determining the superior approach among data-driven, knowledge-driven, aggregated, or interactive methods is challenging due to the variety of data sets and the vast array of potential combinations [41].

## Human-AI collaboration

Human decision-makers receiving advice from algorithmic systems is not new and has been studied for many decades [50]. With AI systems' increasing power and practicality, it has found their way into more and more domains, often surpassing human judgment, even with simple methods [51,52]. While they are not infallible, relying solely on them might yield better results when human decision-making is generally less accurate. Yet, this approach will still fall short of the optimal scenario where human and AI decision-making are complementary [53,54].

Despite the potential benefits of incorporating algorithmic advice in decision-making processes, many individuals reject such recommendations [10,55], leading to an under-reliance on the advice and, therefore, often to a decreased decision-making performance [56]. The phenomenon of advice aversion has been extensively studied in human-to-human interactions [57] and, more recently, between humans and AI [7,8]. Algorithm aversion, as defined by Mahmud et al. [8], refers to neglecting algorithmic decisions in favor of one's own decisions or those of others, consciously or unconsciously. The antithesis of algorithm aversion is algorithm appreciation and automation bias [11], potentially causing decision-makers to over-rely on algorithmic advice. This divergence between aversion and appreciation could be partly attributed to the task's nature. Factors such as whether the task appears more objective or subjective from a human perspective [10], or if the employment of algorithms aligns with prevailing social norms [58], may play significant roles. Recent studies have explored methods to mitigate of over- and under-reliance, such as employing cognitive-forcing functions [59] and providing XAI explanations [54] with mixed results. For an overview of empirical work on human-AI decision-making, we recommend a recent review by Lai et al. [60].

In this regard, we adopt the definition of reliance provided by Scharowski et al. [61], which describe it as "*a user's behavior that follows from the advice of the system*". We emphasize that we are not concerned with whether the reliance is *appropriate* or not: In contexts where humans receive advice from AI, decision-making performance can surpass that of individuals only when the human accurately discerns and adheres to correct advice while disregarding erroneous suggestions [53]. Our study's objective is not to enhance the performance of AI-assisted decision-making by optimizing or calibrating the decision makers' reliance or trust [62]. Instead, we view feature selection as a potential factor influencing reliance that could be considered in optimizing advice-giving systems.

To better understand the factors influencing advice-taking interactions between humans and AI, numerous studies have investigated the effects of different AI aspects and advice-taker characteristics. Sundar [19], in his framework for studying human-AI interactions, argues that AI elements can serve as cues that trigger cognitive heuristics during an interaction. These heuristics, which he refers to as "machine heuristics," can be perceived positively or negatively and depend on individual differences [63]. In their review, Mahmud et al. [8] group influencing factors into four categories: task factors (e.g., subjectivity and morality), high-level factors (e.g., social norms), individual factors (e.g., fear of change, expertise, and demographics), and algorithmic factors (e.g., explainability, accuracy, and integration). Jussupow et al. [7] similarly categorize factors into algorithm characteristics (agency, performance, capabilities, and human involvement) and human agent characteristics (social distance and expertise). Our study focuses explicitly on the feature selection method as a factor. This process is categorized under algorithmic factors and characteristics. It is also related to the category of human involvement in AI systems. In our case, this involves integrating humans as experts and decision-makers in the feature selection process and also the later interaction between decision-maker and AI.

Jussupow et al. [7] emphasize distinguishing who is involved in the machine learning pipeline, whether it is the later end-user or a human developer (e.g., a data scientist) integrated into the development process. Experiments by Jago [21] demonstrate that expert involvement in the training process can enhance algorithm authenticity. Interestingly, participants tend to prefer models trained by data scientists over purely automated methods, as observed by Ashoori and Weisz [9], and they do not even differentiate between prestigious and non-prestigious institutional affiliations [64]. Palmeira and Spassova [65] found that people prefer a combination of expert judgment and decision aid over expert judgment alone. Their results are similar to Waddell's [20], who investigated the differences in the perception of human and algorithmic authors of journalistic articles and found that biases are attenuated when humans and algorithms work in tandem. Lastly, Cheng and Chouldechova [22] investigate three ways in which humans can control AI decisions: altering the input, controlling the process (e.g., the learning algorithm), and adjusting the output for the final decision (the most common type of control in the literature). They found that process and output control reduce algorithm aversion while input modification does not.

Literature exploring algorithm appreciation and aversion suggests that decision-makers favor human involvement in the machine learning process and that human involvement decreases algorithm aversion. Consequently, we hypothesize that when given a choice, users of machine learning models are more inclined to prefer an machine learning model that uses features selected by experts rather then by an algorithm.

**H1a:** *A expert feature selection method is chosen more frequently than a algorithmic feature selection method.*

A machine learning model that uses a combination of an expert and algorithm feature selection method can be perceived as a "tandem," similar to what Waddell's study showed about the joint effort of algorithms and humans [20]. The involvement of two parties in this process may lead to a cumulative [18] or a "double-dose" effect [66]. Echoing Palmeira's and Spassova's [65] findings, which suggest a preference for combined efforts over sole expert judgment, we hypothesize that the model utilizing a combined method will be more favored than the expert method. Furthermore, we believe that its advice will likely garner the highest level of reliance.

**H1b:** *A combination of expert and algorithmic feature selection methods is chosen more frequently than an expert feature selection method alone.*

We also think that these preferences can be transferred to reliance, allowing us to formulate hypotheses accordingly:

**H2a:** *Advice generated using an expert feature selection method exhibits higher reliance rates than those generated with an algorithmic feature selection method.*

**H2b:** *Advice generated using a combination of expert and algorithmic feature selection methods exhibit higher reliance rates than those generated with an expert feature selection method alone.*

We excluded a variety of feature selection methods here, as we are primarily focused on the different levels of human involvement, and thus concentrate on three distinct stages.

Permitting user to choose their preferred feature selection method introduces a form of control akin to the experiments conducted by Cheng and Chouldechova [22]. Although their results suggest that allowing decision-makers to control the process should increase reliance, feature selection only influences the input, not the processing of information, which may not affect reliance. Kawaguchi [67] found that workers were more receptive to advice when their predictions were considered. An experiment by Köbis and Mossink [68] found that when participants' opinions were incorporated into the decision-making process, it decreased AI aversion. Burton et al. [69] posit that human-in-the-loop decision-making or even an illusion of autonomy can mitigate algorithm aversion. Other factors may explain why the participant's choice might influence reliance positively. For example, the sunk cost fallacy suggests that participants who have invested time and effort in choosing a feature selection method may be more inclined to rely on the model's predictions to justify their initial choice [70].

**H3:** *Giving the users choice to choose their prefered feature selection method positively increases the reliance on the machine learning model's advice.*

## Methods

We employ a behavioral experiment with a between-subject design and two treatments. Our experimental design draws inspiration from prior research on human-AI decision-making processes [60]. It incorporates two distinct decision-making domains: *Cardio*, which focuses on medical diagnoses, and *Football*, which centers around estimating soccer match outcomes. In the first treatment *Choice*, we investigate the decision-maker's preference for these methods when given a choice. Second, we compare this group with another treatment group *No Choice*, which had no option to choose their preferred method. The *No Choice* treatment has three sub-treatments: a human selects features, a data-driven algorithm selects, or feature selection results from a joint effort. We assess the decision-maker's reliance on algorithmic advice in all settings. Do people also prefer ex-ante to what they will rely on ex-post?

Moreover, in an exploratory manner, we examine the correlation between the characteristics of decision-makers and their preferences and reliance on advice. By identifying personality traits related to preference and reliance, we aim to augment the existing literature that has predominantly centered on general trust and reliance rather than specific aspects like feature selection [8,29,71,72]. Hyperlinks to the experimental data can be found in Data within S1 Data.

### Participants and treatments

**Participants.** A total of 265 participants were recruited from Prolific.com between August 2nd and 18th, 2023. The participants were informed about the study and data protection before the start of the experiments and gave their consent digitally; otherwise, they could not participate. The Paderborn University Institutional Review Board approved the study as part of the research project. Each participant provided voluntary and digital consent before the

start of the experiment. Initially, 16 participants were excluded due to failing an initial comprehension check, while another 29 withdrew. Additionally, 4 participants were removed after failing attention checks. Consequently, the final sample comprised 216 participants for analysis. 129 (59.7%) were women, and the average age was 34.2. Participants required, on average, 27.3 minutes to finish the study and earned an average payment of £9.63. We exclusively recruited participants from the United Kingdom to ensure English language proficiency and a higher likelihood of a basic understanding of football, one of the task domains. Upon completing the study, participants received a fixed payment of £5. Additionally, participants received bonus payments contingent upon the accuracy of their decisions.

**Treatments.** 109 participants were randomly assigned to the *Choice* treatment. In this treatment, participants determined who would be responsible for selecting the features upon which the advising AI is trained for both task domains. The remaining 107 participants were assigned to the *No Choice* treatment. Unlike the other treatments, they were not given a choice between methods; instead, they were randomly allocated to one.

## Experimental procedure

The experimental software for this study was developed using oTree [73] and was deployed online. Participants were required to access the study through a desktop client to minimize the risk of distractions and technical issues. The experiment itself is an incentivized behavioral experiment that adheres to design principles found in related literature [60,74,75].

The study began with an explanation of the data protection policy, followed by the general instructions for the study (see Instructions in S1 Text). Participants were then presented with multiple comprehension questions, with a maximum allowance of two incorrect responses for each question.

The main component of the study is the experiment, including the classification tasks and an advice-giving AI. Screenshots of the classification task and advice-giving can be found in S1 Fig and S2 Fig. Participants were asked to perform multiple binary classification tasks, wherein they provided with information on decision problems and required to submit answers. Participants were awarded additionally £0.20 for each correctly solved task. Upon completion, participants completed a survey to collect demographic and personality information.

**Judge-advisor system.** A Judge-Advisor System (JAS), commonly employed in advice-taking research, was utilized in the experiment [57]. Within the JAS, the participant (acting as the decision-maker) is presented with a decision problem. The participant makes an initial decision based on the information provided for the problem. After submitting this initial decision, an advisor (in this case, a machine learning model) offers advice. The participant then makes a subsequent decision, allowing them to reconsider and possibly modify their initial decision by incorporating the advice as they see fit. Moreover, for each initial decision, participants were prompted to rate their confidence on a slider input ranging from 0 (absolutely not confident) to 100 (very confident), with the default value set to 0 [76]. It is central to note that the decision and the advice are presented on the same scale. Screenshots of the decision pages can be found in the S1 Fig and S2 Fig.

A subtle but important distinction between our study and many prior studies in the JAS literature is that advice was provided only when they deviated from the initial decision. In other JAS experiments, the decision problems often involve regression tasks with cardinal answers, making it more likely for discrepancies between the participant's decision and the advice. However, since our study focuses on binary decisions, offering advice that aligns with

the initial decision seems redundant and offers little to no insight [53]. In a pre-study involving ten students, we observed that when their initial decision matched the advice, an alternation of the participants' decisions did not happen. This appears quite logical: typically, one would only diverge from the advice (that mirrors their own belief) if there's a firm conviction of its inaccuracy. Omitting advice when the advice would only confirm the respondents' initial choice was more efficient. Participants learned they would only revive advice when their initial choice and that AI recommendations would diverge. Participants were briefed about this approach in the instructions.

## Classification domains and machine learning models

**Domains and tasks.**   To guarantee the generalizability of our study and reduce the influence of domain-specific effects, we utilized two distinct domains for the decision problem tasks that participants performed during the experiment. These two problems, labeled as and are derived from publicly available datasets.

The *Cardio* problem is a classification task that involves predicting the presence of cardiovascular disease using patient characteristics and symptoms. The dataset for this problem consists of 70,000 patients. The second classification problem, *Football*, focuses on determining whether the home team in a football match won or not, based on match statistics. The original dataset contains 4,070 matches.

These datasets were selected carefully to ensure comprehensibility for the experiment's participants regarding the decision problem and the incorporated features. Furthermore, we sought a diverse set of domains to avoid domain-specific results, as the domain can influence advice reliance due to different task-related factors. For instance, humans exhibit higher aversion for tasks perceived as more subjective than objective [10,77] or when facing morally relevant decisions, particularly in legal or medical fields [78].

We opted for 20 tasks for each domain to allow participants to become more familiar with the decision problem and experience multiple advice-receiving instances. Previous studies have observed that algorithm aversion tends to weaken over time [79]; thus, incorporating multiple tasks should enhance the reliability of our results. Participants were neither provided with feedback about the correctness of their decisions between rounds nor the accuracy of the ML models. This was an intentional choice to focus on the immediate effects of feature selection methods on user preferences and reliance without introducing additional variables that could influence behavior. Providing immediate feedback could lead participants to adjust their strategies based on performance outcomes, potentially introducing noise and confounding the specific effects we aimed to measure. Instead, they received information about their overall payment only at the end of the study.

**Feature subsets.**   To maintain comparability between domains, it was necessary to standardize the number of features employed in both the tasks and the models across all three decision problems. Moreover, we needed to provide the models and the participants with sufficient information to make useful predictions. A vital design aspect of the experiment was to explain to participants that a selection of features had occurred and that a selection could impact the quality of the advice. Participants were given 12 features for solving the classification tasks in each decision problem. Still, only 6 of the 12 features were used for the ML models, which were shown and highlighted to the participants. We believe using a subset of the features renders the selection process more intelligible and pertinent. Although supplying participants with more information than the models might adversely affect advice reliance, we also contend that decision-makers in many real-life situations possess a different set of information that could contain more detail.

During the experiment, to ensure that all treatments were equal in all aspects except the feature selection method, it was also vital that the features used for predictions remained consistent in all selection methods, guaranteeing that the advice was uniform across all treatments. We carefully selected the final feature sets employed in the task using multiple feature selection algorithms. For the two domains, we selected the following features, with the first 6 in the list being used for the machine learning models:

*Cardio*: Age, Weight in kg, Body Mass Index, Systolic blood pressure, Diastolic blood pressure, Cholesterol level, Gender, Height in cm, Glucose level, Smoking status, Alcoholism, Physical activity.

*Football:* Offsides away team, Passes away team, Passes home team, Possession home team in %, Shots away team, Shots home team, Corners away team, Corners home team, Fouls conceded home team, Offsides home team, Yellow cards away team, Yellow cards home team.

**Machine learning model.** To train the ML models responsible for the advice, we employed the XGBoost algorithm, a widely used and highly effective algorithm for classification and regression tasks [80]. To ensure the optimal performance of our models, we performed model tuning using the grid search method in conjunction with 5-fold cross-validation. We divided each dataset into a training and a test set. The training set was utilized for hyperparameter tuning and learning, while the test set was employed for evaluating the model's performance. We evaluated the final models using balanced accuracy. The *Cardio* model scored 0.74, while the *Football* model scored 0.64. Although these scores are not exceptionally high and might be considered insufficient for practical applications, their impact on the experiment is likely minimal, as the participants were not briefed on the models' performance. For the tasks, we selected observations, ensuring that the model's accuracy for these specific observations was roughly equivalent to its performance on the test dataset. The sequence of the two domains and the order of tasks were randomized for each participant.

## Evaluation measures

**Advice reliance measurement.** In our study, we primarily aim to explore participants' preferences for the feature selection method and how these methods influence their reliance on the advice. Hereto, we adopt the approach used in two recent studies [53,75]. As the judgments and advice in these tasks are binary (e.g., no disease/disease, home team won/home team did not win), we are particularly interested in instances where the participant's initial decision is unequal to the model's advice. Observing how the participant reconciles the conflicting answers is interesting in such cases. If the participant alters their belief in the subsequent decision to align with the advice rather than maintaining their initial decision, we consider this a reliance on advice. Consequently, the dependent variable is referred to as *Switch to Advice*.

**Explanatory variables.** We draw upon established scales from various social science disciplines to measure individual characteristics. The Big Five personality traits (Openness, Conscientiousness, Extraversion, Agreeableness, and Neuroticism) are measured using ten items on a 5-point Likert scale [81]. The lottery choice task by Gächter et al. [82] measures loss aversion. For risk-taking, we rely on the Global Preference Survey (GPS) by Falk et al. [83], which uses a scale and multiple preference-related questions. We adopt two scales to measure affinity for technology (ATI) [84] and artificial intelligence (GAAIS) [72]. ATI consists of 9 items on a 6-point Likert scale. At the same time, GAAIS is divided into two dimensions—positive affinity, measured with 12 items, and negative affinity, assessed through 8 items—both using a 5-point Likert scale.

## Results

The analysis is segmented into two main sections. In the first section, we initially examine the feature selection methods chosen by participants in the *Choice* treatment. The primary aim is to test the first two hypotheses: Do individuals prefer *Expert* over *Algorithm*, and is *Combination* the most favored? Additionally, we seek to determine if distinctions exist between the two domains. In the explanatory segment of this section, we delve into the participant characteristics associated with their choices.

In the second section, we address three hypotheses concerning advice reliance—do individuals' ex-ante preferences align with what they end up relying on ex-post? The dependent variable in this section is *Switch to Advice*, which denotes instances when participants amend their subsequent decisions to the AI's prediction when the advice diverges from their initial decision. We will consider both the participants of the *No Choice* and the *Choice* treatments. This will allow us to determine if choosing the methods influences advice reliance for the third hypothesis. In the explanatory segment of this section, we explore the participant characteristics associated with reliance.

### Feature selection preferences

**General preferences.** During the *Choice* treatment ($N = 109$ participants with two decisions resulting in $n = 218$) the feature selection method *Algorithm* was chosen 44 times (20.2%), *Expert* 70 times (32.1%), and *Combination* 104 times (47.7%). The chi-squared test indicates that this distribution significantly deviates from what would be expected in a random sample ($\chi^2 = 24.917, P < 0.001$). Pairwise comparisons reveal significant distinctions among all three methods: *Algorithm* vs. *Combination* ($\chi^2 = 23.324, P < 0.001$), *Algorithm* vs. *Expert* ($\chi^2 = 5.93, P = 0.015$), and *Combination* vs. *Expert* ($\chi^2 = 6.644, P = 0.001$). Fig 1 illustrates the distribution of the selections.

**Preferences between domains.** Based on these findings, one might accept hypotheses 1a and 1b, which posit that *Expert* is preferred over *Algorithm* and that *Combination* is favored over *Expert*. However, when examining the data segregated by domains, it becomes evident that participants' preferences are more nuanced and not as straightforward. In *Cardio*, *Algorithm* was chosen 18 times (16.5%), *Combination* 51 times (46.8%), and *Expert* 40 times (36.7%). Once more, we note that the distribution significantly deviates from that of a random sample ($\chi^2 = 15.541, P < 0.001$). Unlike in the analyses conducted on the entire dataset, the pairwise comparison reveals that the difference between *Combination* and *Expert* is no longer significant ($\chi^2 = 1.33, P = 0.25$). Still, the differences between *Algorithm* and both *Combination* ($\chi^2 = 15.783, P < 0.001$) and *Expert* ($\chi^2 = 8.345, P = 0.004$) are statistically significant. In *Football*, a distinct pattern is observed: *Algorithm* was chosen 26 times (23.9%), *Combination* 53 times (48.6%), and *Expert* 30 times (27.5%). Once again, the distribution significantly diverges from that of a random sample ($\chi^2 = 11.688, P = 0.003$). *Combination* was significantly more favored compared to both *Algorithm* ($\chi^2 = 9.228, P = 0.002$) and *Expert* ($\chi^2 = 6.373, P = 0.003$), but no significant difference is found between *Algorithm* and *Expert* ($\chi^2 = 0.285, P = 0.593$). Fig 2 illustrates the selection distributions for both domains. To determine if participants' first and second choices were independent, we examined the distribution of preferences for these choices. Our comparison showed no significant differences ($\chi^2 = 2.138, P = 0.343$). This independence in preferences was observed irrespective of whether *Cardio* ($\chi^2 = 4.092, P = 0.129$) or *Football* ($\chi^2 = 1.561, P = 0.458$) was the first domain in the experiment. While the general analysis allows us to accept both hypotheses H1a and H1b, we point to domain-specific differences that influence the relationships.

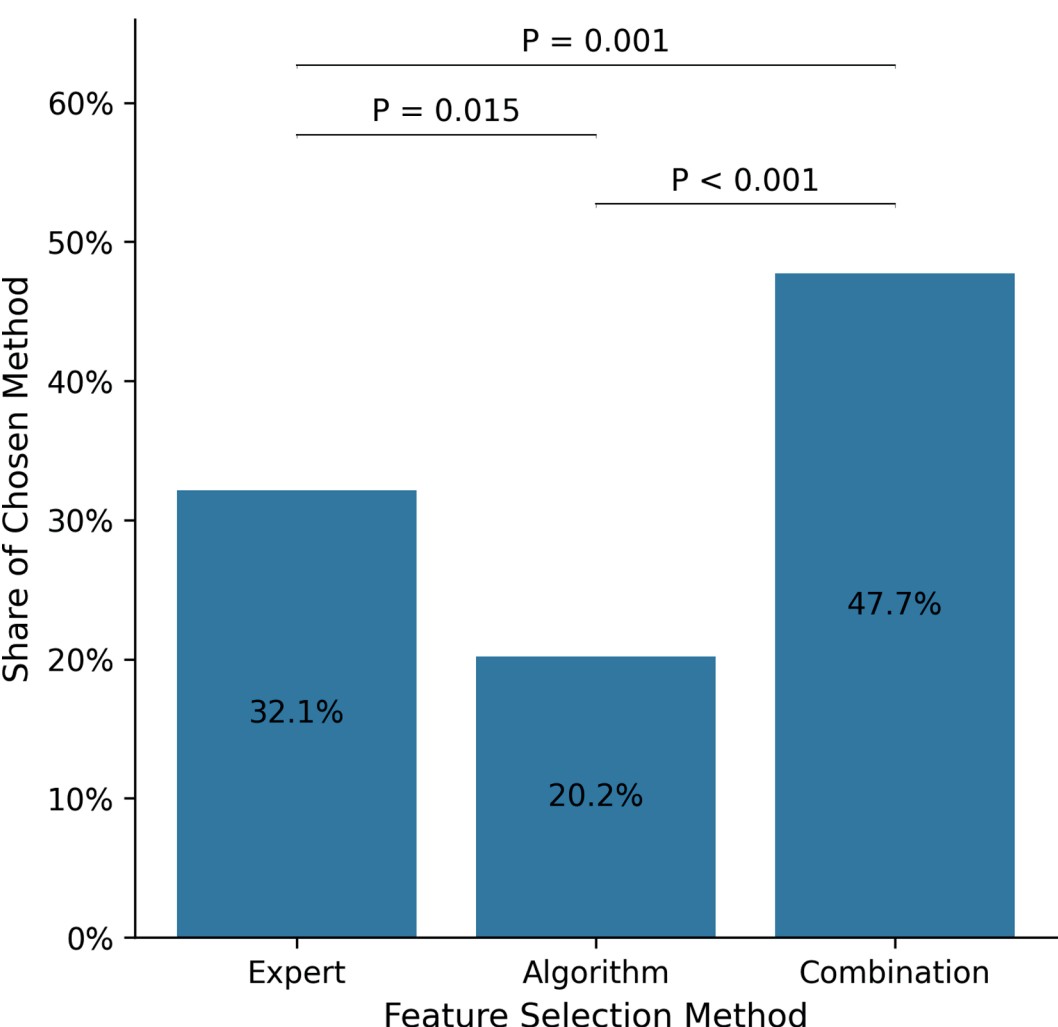

**Fig 1. Distribution of the chosen feature selection methods.**

**Exploration of characteristics.** Regarding personality characteristics, we found using two multinomial logistic regression models (Table 1) that age is negatively associated with a preference for *Expert* when compared to *Algorithm* ($\beta = 0.038, SE = 0.02, P = 0.06$) and *Combination.* ($\beta = 0.032, SE = 0.017, P = 0.06$). *Neuroticism* is positively associated with an increased preference for *Combination* when compared to *Expert* ($\beta = 0.469, SE = 0.233, P = 0.045$) and *Combination* to *Algorithm* ($\beta = 0.754, SE = 0.264, P = 0.004$). *Risk-taking* is positively linked with an augmented preference for both *Algorithm* ($\beta = 1.616, SE = 0.687, P = 0.018$) and *Combination* ($\beta = 1.458, SE = 0.557, P = 0.009$) over *Expert*.

## Advice reliance

**Descriptive statistics.** In contrast to the previous section, we now utilize data from both treatments, so we observe 216 participants from *Choice* and *No Choice* together. The machine learning models outperformed the participants in the classification tasks. Their predictions were correct in 65% of the *Cardio* and in 60% in *Football* tasks. Participants initially decided correctly in 54.69% of cases (*Cardio*: 63.40%, *Football*: 46.37%). The initial decision aligned

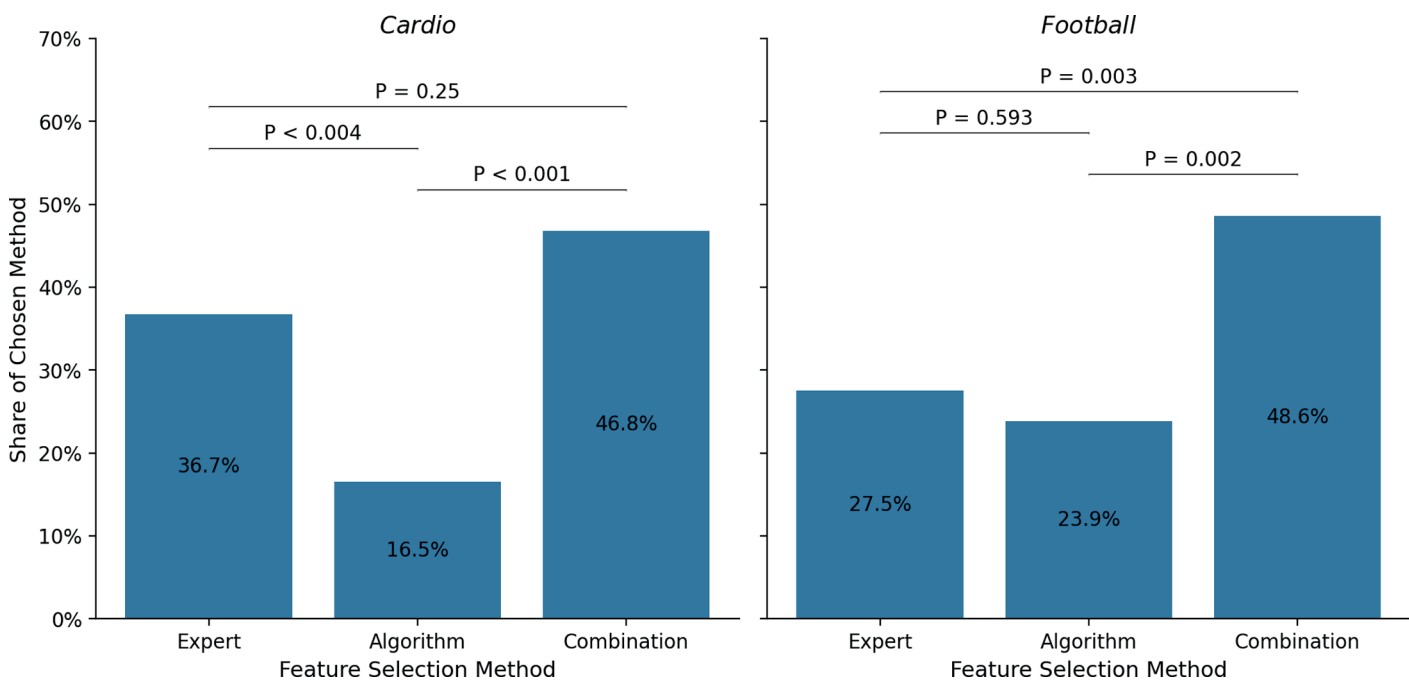

**Fig 2. Distribution of the chosen feature selection methods for both domains.**

**Table 1. Multinomial Logistic Regression results for the feature selection method preferences.**

| Base Category | Expert | | Algorithm | |
|---|---|---|---|---|
| | Algorithm | Combination | Combination | Expert |
| *Cardio* | −0.714† (0.408) | −0.358 (0.325) | 0.356 (0.378) | 0.714† (0.408) |
| Male | −1.051* (0.500) | −0.485 (0.396) | 0.566 (0.456) | 1.051* (0.500) |
| Age | 0.038† (0.020) | 0.032† (0.017) | −0.006 (0.018) | −0.038† (0.020) |
| Big 5 Extraversion | −0.004 (0.245) | −0.043 (0.189) | −0.038 (0.232) | 0.004 (0.245) |
| Big 5 Agreeableness | −0.105 (0.316) | −0.221 (0.249) | −0.116 (0.279) | 0.105 (0.316) |
| Big 5 Conscientious-ness | −0.334 (0.293) | 0.039 (0.227) | 0.373 (0.274) | 0.334 (0.293) |
| Big 5 Neuroticism | −0.288 (0.288) | 0.466* (0.233) | 0.754** (0.264) | 0.288 (0.288) |
| Big 5 Openness | 0.032 (0.248) | −0.103 (0.199) | −0.135 (0.225) | −0.032 (0.248) |
| Loss Aversion | −0.137 (0.150) | −0.095 (0.124) | 0.042 (0.137) | 0.137 (0.150) |
| Risk Taking | 1.619* (0.687) | 1.458 (0.557)** | −0.161 (0.620) | −1.619* (0.687) |
| ATI | 0.221 (0.267) | 0.085 (0.204) | −0.137 (0.243) | −0.221 (0.267) |
| GAAIS Positive | 0.278 (0.371) | 0.357 (0.296) | 0.079 (0.353) | −0.278 (0.371) |
| GAAIS Negative | 0.391 (0.338) | 0.053 (0.264) | −0.338 (0.308) | −0.391 (0.338) |
| *n* (Choices) | 218 | | | |
| *N* (Participants) | 109 | | | |
| Pseudo $R^2$ | 0.0812 | | | |

The first two models use *Expert* as their base category, while the third and fourth use *Algorithm*. Standard errors in parentheses. † P<0.1, * P<0.05, ** P<0.1, *** P<0.01.

with the models's prediction in 69.11% of instances (*Cardio*: 73.22%, *Football*: 65.00%). In scenarios where the initial decision did not align with the models's advice, participants were correct 37.69% of the time (*Cardio*: 47.02%, *Football*: 30.55%). Conversely, the models's advice was accurate 62.31% of the time in these situations (*Cardio*: 52.98%, *Football*: 69.44%). Participants chose to switch their decisions to follow the models's advice in 44.77% of these cases

(*Cardio*: 53.93%, *Football*: 37.77%). As a result, the overall accuracy rate in advice-receiving situations amounted to 47.47% (*Cardio*: 49.96%, *Football*: 45.57%).

**Reliance between methods and treatments.** While these results indicate that participants partially rejected the advice and, therefore, exhibited an aversion, it's necessary for our research question to examine how reliance depends on the underlying feature selection method and the participant's choice. Fig 3 shows the distribution of *Switch to Advice* across the three methods, distinguishing between both treatments, *Choice* and *No Choice*. Additionally, Fig 4 segregates the data further, delineating the results for both domains.

We employ mixed-effects logistic regression models (Table 2) to analyze whether the methods influence reliance. The regressions incorporate a random intercept for each participant, accounting for the multiple observations per individual. For the pairwise comparisons, we alternately set *Expert* and *Algorithm* as the reference categories. We include a dummy variable for the *Choice* treatment and the *Cardio* domain, the number of rounds, the self-reported confidence in the initial decision, and variables representing participant characteristics.

We note 2,669 instances where participants received advice from the AI, as advice was provided only when they deviated from the initial decision of the participants. Both models demonstrate that the respective methods do not have a significant effect on reliance. Furthermore, the option to choose a method also has no influence. Therefore, we reject the hypotheses H2a, H2b, and H3.

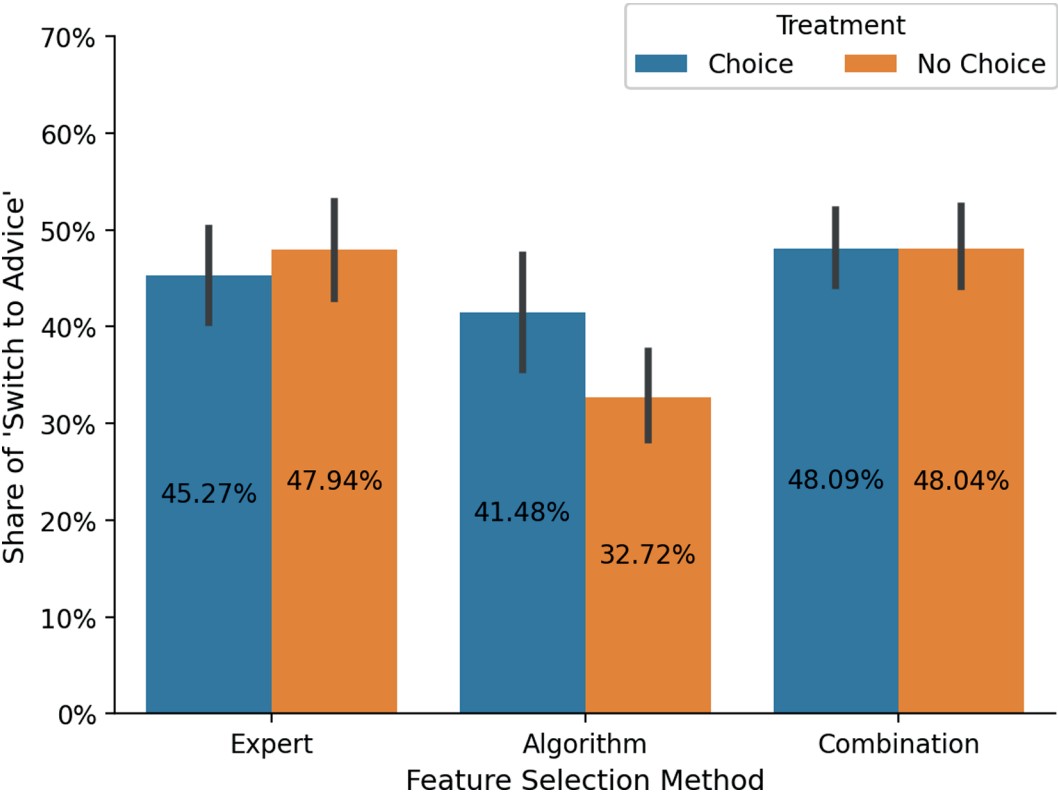

**Fig 3. Distribution of *Switch to Advice* by feature selection methods.** Error bars represent 95% confidence intervals.

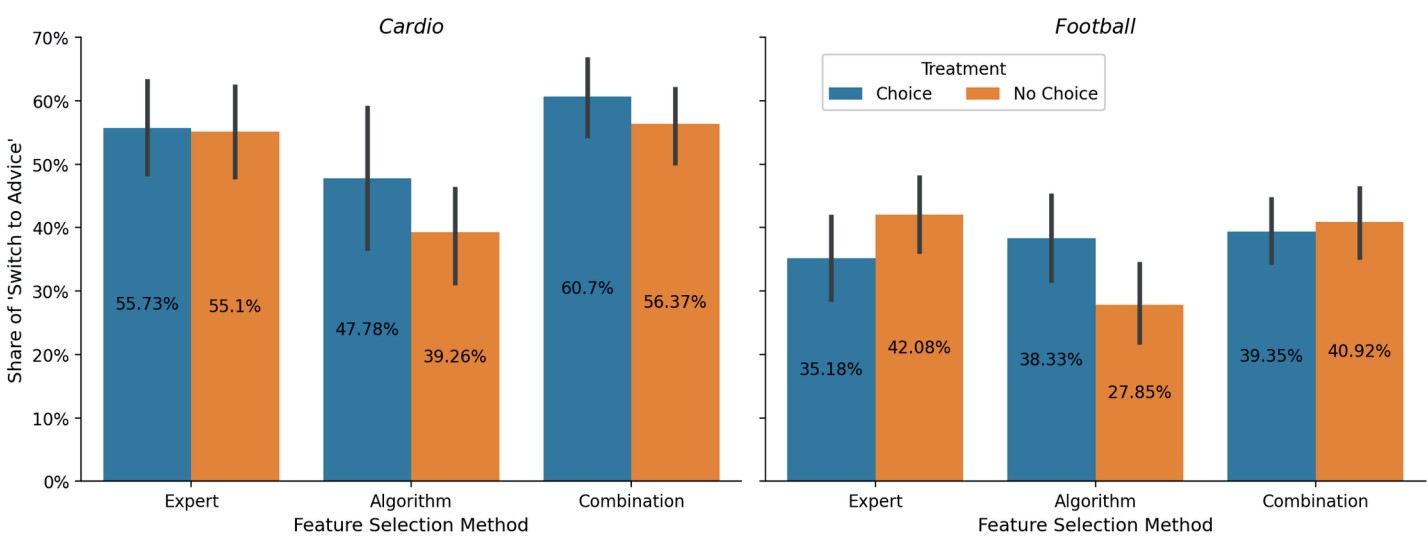

**Fig 4. Distribution of *Switch to Advice* by feature selection methods and domains.** Error bars represent 95% confidence intervals.

A significant domain effect is evident through a significant positive coefficient for *Cardio* ($\beta$ = 1.008, $SE$ = 0.099, $P < 0.001$), a pattern also reflected in our descriptive analysis. This corresponds to a marginal effect of 17.98 percentage points.

**Analyis of covariates.** As the coefficient for the number of tasks is also insignificant, we don't observe any time trends. This was expected as the participants had no feedback during the task. A notable association exists between participants' self-reported confidence in their initial decision and advice reliance ($\beta$ = –0.028, $SE$ = 0.004, $P$ = 0.000). As confidence in one's decision diminishes, the reliance on the AI's advice grows—for each unit (on a scale from 0 to 100), the likelihood of change in the subsequent decision falls by 0.49 percentage points. Regarding personality and demographic attributes, we do not observe any gender-specific effects. However, a significant negative relationship emerges between age and advice reliance ($\beta$ = –0.020, $SE$ = 0.008, $P$ = 0.017). Each year, the likelihood of advice reliance decreases by 0.36 percentage points. Among the Big 5 personality traits, *Openness* is a negative association ($\beta$ = –0.225, $SE$ = 0.107, $P$ = 0.035).

## Discussion

### Main findings

To begin with, we discover that decision-makers in our experiment prefer the *Expert* over *Algorithm* and favor *Combination* over *Expert*. Yet, when separating the data by the two domains, it becomes evident that the specific domains may have affected participants' choices. In the domain where participants classified patients based on symptoms and characteristics into groups with and without cardiovascular disease, we find no significant difference between the popularity of *Combination* and *Expert*. In contrast, in determining a home team win based on match statistics, *Combination* is significantly the most popular, with *Algorithm* and *Expert* being equally favored.

In our analysis regarding the classification tasks, we observe, contrary to our expectations, no significant effect of the underlying feature selection methods on advice reliance and no effect of the opportunity to choose the method by the participants. Significant predictors of reliance are the domain (with a higher reliance in the medical domain), personal confidence

**Table 2. Mixed-effects logistic regression results for *Switch to Advice*.**

| | (1) Switch to Advice | (2) Switch to Advice |
|---|---|---|
| *Expert* | / | 0.236 (0.199) |
| *Algorithm* | −0.236 (0.199) | / |
| *Combination* | −0.017 (0.164) | 0.220 (0.189) |
| *Choice* | 0.154 (0.188) | |
| *Cardio* | 1.008*** (0.099) | |
| Round Number | −0.003 (.004) | |
| Own Confidence | −0.028*** (0.003) | |
| Male | −0.292 (0.222) | |
| Age | −0.020* (0.008) | |
| Big 5 Extraversion | −0.065 (0.104) | |
| Big 5 Agreeableness | 0.174 (0.103) | |
| Big 5 Conscientiousness | 0.202† (0.122) | |
| Big 5 Neuroticism | −0.028 (0.116) | |
| Big 5 Openness | −0.225* (0.107) | |
| Loss Aversion | −0.001(0.070) | |
| Risk Taking | 0.203 (0.28) | |
| ATI | −0.042 (0.120) | |
| GAAIS Positive | 0.232 (0.165) | |
| GAAIS Negative | −0.028 (0.143) | |
| Participant Intercept | 1.329 (.208) | |
| Constant | 0.626 (1.253) | 0.556 (1.253) |
| Log-likelihood | −1565.109 | |
| Wald $\chi^2(23)$ | 185.89 | |
| Prob > $\chi^2$ | 0.000 | |
| LR test vs. logistic model: $\overline{\chi}^2(01)$ | 270.53 | |
| Prob ≥ $\overline{\chi}^2$ | 0.000 | |
| Observations | 2,669 | |
| Number of groups | 216 | |

The first model uses *Expert* as the base category, and the second *Algorithm*. Standard errors in parentheses. † P<0.1, * P<0.05, ** P<0.1, *** P<0.01.

in the decision, and age, both showing negative correlations with reliance. From the Big 5 scale *Openness* was negatively associated with reliance.

Together, the findings from our analysis of preferences do not align with those concerning reliance. Given the notable differences in popularity between *Combination* and both *Algorithm* and *Expert* (especially in one domain), one might anticipate greater advice reliance on *Combination* during the classification task. Yet, we observe no effect. While AI users express their preferences regarding AI characteristics, their ultimate behaviors remain largely uninfluenced by these stated preferences. This result is similar to two previous studies: Rabinovitch et al. [28] found that participants explicitly preferred a human advisor over an algorithmic one, but the advice was used equally. Rebitschek et al. [29] discovered a discrepancy between the acceptable, perceived, and actual error rates of algorithms. This can be attributed to various cognitive factors. For instance, according to dual-process theory [85], when asked about their preferences, participants may have engaged in System 2 thinking, carefully evaluating the perceived benefits of the three options. However, during the actual decision-making process, they likely reverted to System 1 thinking due to the complexity of the task and the cognitive load. As a result, they may have paid less attention to the subtle details of the feature selection methods. Another possible explanation is social desirability bias [86], which

could have led participants to perceive the combined feature selection method as the most advanced, and therefore, the most acceptable option.

In conjunction with the unobserved selection effect, these results resonate with the findings of Cheng and Chouldechova [22]. Their research suggested that while choosing the training algorithm can alleviate algorithm aversion, modifications to the information utilized by the algorithm do not offer similar mitigation. Our results partly confirm the framework by Jussupow et al. [7], as in our study, humans state a preference for human involvement in AI development by asking humans to (partly) select the features. However, we find no evidence that this stated preference also unfolds its effects when humans face AI advice. Gogoll and Uhl [87] found a comparable trend: while their participants leaned towards delegating tasks to humans over machines, their trust did not differ.

## Secondary findings

In addition to the relationships of the treatments analyzed, our results indicate that other factors, notably the task domain and the users themselves, play a significant role. Our results indicate caution when analyzing human-AI collaborations, as results may be artifact-specific. Utilizing a self-reported scale for risk-taking behavior [83], a multinomial model shows that participants displaying higher risk-taking tendencies exhibited a preference for *Algorithm* and *Combination* over *Expert*. This inclination might be explained by the "Diffusion of Innovations" theory—historically, early adopters of novel technologies tend to be more risk-prone [88,89]. If *Expert* is perceived as more conservative, then a method incorporating or entirely based on algorithms might be perceived as a more innovative approach.

We observe a significant positive effect of the medical domain on the likelihood of adjusting the decision toward the AI prediction. Notably, our findings do not entirely align with previous research on algorithm aversion in medical settings. For instance, Arkes and Blumer [70] reported that participants favored physicians who did not utilize decision aids. Similarly, Longoni et al. [90] noted a hesitancy towards AI providers compared to human providers in a medical context. While reliance is typically linked to perceived risk, and medical decisions usually carry more risk than sports-related ones, the payoff for both domains is identical, making the risk equivalent. Other factors contributing to the differences in reliance could include perceived AI competence in each domain or participants' own confidence in their classification abilities. However, in this case, we observed higher confidence among participants in the medical domain. Our analysis indicates a significant negative correlation between the decision-makers' confidence and their reliance on AI, consistent with prior experimental findings [11,56,91]. The inverse relationship between a participant's age and reliance diverges from findings by Ho et al. [92], who determined that older adults exhibited a higher trust in decision aids. Similarly, Logg et al. [11] discovered a consistent appreciation for algorithms irrespective of age. Gender was not a significant predictor, as in the study by Logg et al. [11]. The reported inconsistencies may be partially attributed to the rapid integration of AI into society. This is because algorithm aversion and appreciation can be understood through normative processes [58] and long-term learning effects [79].

## Limitations and implications

One potential reason for the missing differences in reliance between the methods might be due to a manipulation that is too subtle. There's a possibility that the methods' signals are too faint within the task to detect an effect corresponding to the significant differences observed in preferences. Despite this, the presentation mirrors real-world scenarios where detailed explanations of AI feature selection methods are rarely provided. Participants were able to

review the selected features during the tasks, unlike during the method selection phase. This visibility allowed them to reasonably assess the selection's validity, likely comparing it with their judgment. Consequently, the feature selection method information likely serves as only a minor indicator of the selection's validity, possibly leading to the observed results. Future studies might consider not displaying the features, although this approach could reduce realism.

Another limitation impacting the generalizability of our findings is the recruitment of non-professional decision-makers from an online participant pool instead of domain professionals. While we acknowledge that expertise is crucial in many real-world applications, using lay participants offers important advantages, especially in the context of fundamental research like ours. Lay participants provide an opportunity to study baseline human-AI interactions without the influence of pre-existing domain-specific knowledge, allowing us to isolate general behavioral patterns related to trust, preferences, and reliance on AI systems. Future studies could build on this foundation by replicating the experiment with domain experts to enhance the real-world applicability of our findings.

Nonetheless, it is plausible that domain experts would not yield substantially different outcomes. On the one hand, the literature reveals that the same biases are prevalent among both laypeople and experts [93,94]. On the other hand, a meta-analysis shows that in human-AI collaboration experiments, there are no differences in decision-making performance between professional and non-professional participants [95]. We believe that, in addition to expertise in one's own domain, experience in machine learning and feature selection is also needed to form a strong opinion. With only domain experience, we expect similar results as seen with laymen, both concerning the preference for human oversight and the reliance on AI advice.

Another way to expand the research in this study would be to shift the focus from short-term interactions to long-term time horizons, exploring how preferences and reliance evolve over time. Long-term research has often been avoided in the human-AI literature due to its empirical challenges, but previous studies suggest the presence of temporal effects [79].

By examining algorithm-based, expert-based, and combined feature selection approaches, we offer fresh insights into how human involvement shapes user trust, preferences, and reliance on AI-driven decisions. Our findings highlight the nuanced and complex relationships between human involvement and user behavior, revealing that the degree of human input can significantly influence perceptions of transparency and trustworthiness, yet these perceptions may not always translate into greater reliance on the system. We reveal a significant attitude-behavior gap, known in many disciplines and for many instances: While humans reveal strong stated preference for human oversight ex ante, individuals are equally likely to rely on AI advice, independent of human oversight.

Our results have practical implications, especially when transparency is essential in decision support systems and there is a lack of trust towards them. Those overseeing or designing AI systems could communicate that the data the AI uses was selected from a joint effort between human experts and algorithms. However, they also need to consider individual traits. As AI systems are often developed in this way, making this known might align with users' preferences, potentially increasing the likelihood of using these systems and leading to better decision-making outcomes.

## Conclusion

AI-supported decision-making is becoming increasingly relevant in everyday contexts, making it essential to understand the factors that influence human-AI interactions. While researchers advocate for greater transparency and explainability, it raises questions about

how users perceive different elements. In this paper, we focus on two critical aspects: human involvement and feature selection, both central to many ML models. Our findings suggest that decision-makers tend to prefer a combination of human and algorithmic feature selection methods. However, we also discovered that neither the methods themselves nor the decision-makers' involvement in choosing these methods significantly influences reliance. These insights underscore the complexity of human-AI interactions and highlight the importance of behavioral experiments in this field of research.

## Supporting information

**S1 Text.**
(PDF)

**S1 Fig.**
(PDF)

**S2 Fig.**
(PDF)

**S1 Data.**
(PDF)

## Author contributions

**Conceptualization:** Jaroslaw Kornowicz, Kirsten Thommes.

**Data curation:** Jaroslaw Kornowicz.

**Formal analysis:** Jaroslaw Kornowicz.

**Funding acquisition:** Kirsten Thommes.

**Investigation:** Jaroslaw Kornowicz.

**Methodology:** Jaroslaw Kornowicz, Kirsten Thommes.

**Project administration:** Jaroslaw Kornowicz, Kirsten Thommes.

**Resources:** Jaroslaw Kornowicz.

**Software:** Jaroslaw Kornowicz, Kirsten Thommes.

**Supervision:** Jaroslaw Kornowicz, Kirsten Thommes.

**Validation:** Jaroslaw Kornowicz, Kirsten Thommes.

**Visualization:** Jaroslaw Kornowicz.

**Writing – original draft:** Jaroslaw Kornowicz, Kirsten Thommes.

**Writing – review & editing:** Jaroslaw Kornowicz, Kirsten Thommes.

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
