## [Decision Letter · Decision Letter 0]

15 Sep 2024

PONE-D-24-30693Algorithm, Expert, or Both? Evaluating the Role of Feature Selection Methods on User Preferences and ReliancePLOS ONE

Dear Dr. Kornowicz,

Thank you for submitting your manuscript to PLOS ONE. After careful consideration, we feel that it has merit but does not fully meet PLOS ONE’s publication criteria as it currently stands. Therefore, we invite you to submit a revised version of the manuscript that addresses the points raised during the review process.

We look forward to receiving your revised manuscript.

Kind regards,

Sohaib Mustafa, Ph.D.

Academic Editor

PLOS ONE

“We gratefully acknowledge funding by the German Research Foundation (Deutsche

Forschungsgemeinschaft, DFG): TRR 318/1 2021 – 438445824.”

“We gratefully acknowledge funding by the German Research Foundation (Deutsche Forschungsgemeinschaft, DFG): TRR 318/1 2021 – 438445824.

“We gratefully acknowledge funding by the German Research Foundation (Deutsche

Forschungsgemeinschaft, DFG): TRR 318/1 2021 – 438445824.”

6. We notice that your supplementary figures are uploaded with the file type 'Figure'. Please amend the file type to 'Supporting Information'. Please ensure that each Supporting Information file has a legend listed in the manuscript after the references list.

Reviewers' comments:

Reviewer's Responses to Questions

**Comments to the Author**

1. Is the manuscript technically sound, and do the data support the conclusions?

Reviewer #1: Yes

Reviewer #2: Yes

2. Has the statistical analysis been performed appropriately and rigorously? 

Reviewer #1: Yes

Reviewer #2: Yes

3. Have the authors made all data underlying the findings in their manuscript fully available?

Reviewer #1: Yes

Reviewer #2: Yes

4. Is the manuscript presented in an intelligible fashion and written in standard English?

Reviewer #1: Yes

Reviewer #2: Yes

5. Review Comments to the Author

Reviewer #1: 1.Very well written paper, over all good work , highly satisfactory analysis but I think scope of improvement in presented approach as the approach needs refinement and still lacks of novelty, it is application of existing approaches , no significant innovative contribution from authors side , we would like to listen in this regard from authors side .

2. These few recent state-of-the-art studies should be discussed where human disease prediction related problems have been thoroughly discussed using feature selection (Feature subset selection through nature inspired computing for efficient glaucoma classification from fundus images;Efficient feature selection based novel clinical decision support system for glaucoma prediction from retinal fundus images;A novel hybrid robust architecture for automatic screening of glaucoma using fundus photos, built on feature selection and machine learning-nature driven computing;An enhanced and efficient approach for feature selection for chronic human disease prediction: A breast cancer study)

Best Regards

Reviewer #2: Paper Title

Algorithm, Expert, or Both? Evaluating the Role of Feature Selection Methods on User Preferences and Reliance

The paper tackles an important and timely topic in the intersection of human-computer interaction and AI transparency. As the demand for explainable AI (XAI) grows, understanding how users prefer to engage with feature selection processes is highly relevant. Moreover, the study contributes to the existing literature by addressing the gap in the interaction between expert-driven and algorithmic feature selection, which has practical implications for AI deployment. Moreover, I have added the following comments to enhance the quality of the work.

Comments:

1. Use of Non-Expert Participants:

o The study utilizes lay participants instead of domain experts, limiting the generalizability of the findings to real-world professional settings where expertise is crucial. Resolve this limitation in the revised manuscript.

2. Limited Scope of Feature Selection Methods:

o The study only considers three feature selection methods (algorithm-based, expert-based, and combined), excluding more advanced techniques like dynamic feature selection or metaheuristic optimization, which could provide a more comprehensive exploration of feature selection strategies. Resolve this limitation in the revised manuscript.

3. Stated Preferences vs. Actual Reliance Discrepancy:

o The study highlights a gap between users' stated preferences and their actual reliance on feature selection methods but does not delve deeply into the cognitive reasons behind this discrepancy.

4. Domain-Specific Results:

o The study shows different results in user reliance between domains (e.g., healthcare vs. sports), but does not thoroughly investigate why domain-specific factors influence these outcomes.

5. Subtle Manipulation of Feature Selection Methods:

o The distinction between the feature selection methods may not have been clear enough to participants, which could dilute the observed effects on decision-making and reliance.

6. Lack of Long-Term User Behavior Analysis:

o The experiment captures short-term interactions with AI and does not assess how user preferences and reliance evolve, missing insights into the long-term dynamics of AI trust.

7. No Real-Time Feedback on Decisions:

o Participants did not receive real-time feedback on their decisions, limiting the understanding of how performance feedback might influence reliance on AI systems.

8. Absence of Task Complexity Analysis:

o The study does not investigate how the complexity of tasks influences user reliance on different feature selection methods, potentially overlooking a key factor that impacts decision-making.

9. Single Decision-Making Context:

o The study uses only two decision-making domains (healthcare and sports), limiting the generalizability of the results across a wider range of industries and decision-making environments.

10. Focus on Binary Decision Tasks:

• The study focuses solely on binary decision tasks, which may not fully capture the complexity of real-world decision-making processes that often involve multiple choices or continuous variables.

11. Discuss the scope and relevance of the following studies in terms of User Preferences and Reliance in your work.

• Tiwari, A., & Chaturvedi, A. (2022). A hybrid feature selection approach based on information theory and dynamic butterfly optimization algorithm for data classification. Expert Systems with Applications, 196, 116621.

• Tiwari, A., & Chaturvedi, A. (2021). A novel channel selection method for BCI classification using dynamic channel relevance. IEEE Access, 9, 126698-126716.

• Tiwari, A., & Chaturvedi, A. (2023). Automatic EEG channel selection for multiclass brain-computer interface classification using multiobjective improved firefly algorithm. Multimedia Tools and Applications, 82(4), 5405-5433.

• Tiwari, A. (2023). A logistic binary Jaya optimization-based channel selection scheme for motor-imagery classification in brain-computer interface. Expert Systems with Applications, 223, 119921.

• Tiwari, A. (2023). A hybrid feature selection method using an improved binary butterfly optimization algorithm and adaptive β–hill climbing. IEEE Access, 11, 93511-93537.

• Yin, L., Wang, L., Lu, S., Wang, R., Ren, H., AlSanad, A.,... Zheng, W. (2024). AFBNet: A Lightweight Adaptive Feature Fusion Module for Super-Resolution Algorithms. Computer Modeling in Engineering & Sciences , 140(3), 2315-2347. doi: https://doi.org/10.32604/cmes.2024.050853

6. PLOS authors have the option to publish the peer review history of their article (what does this mean?). If published, this will include your full peer review and any attached files.

Reviewer #1: No

Reviewer #2: **Yes: **Dr. Anurag Tiwari

---

## [Author Response · Author response to Decision Letter 1]

14 Oct 2024

We would like to thank you for the quick review and the many good improvement suggestions. Our detailed response to the Decision Letter and the description of our improvements can be found in the document "Response to Reviewers".

---

## [Decision Letter · Decision Letter 1]

27 Dec 2024

PONE-D-24-30693R1Algorithm, Expert, or Both? Evaluating the Role of Feature Selection Methods on User Preferences and ReliancePLOS ONE

Dear Dr. Kornowicz,

Thank you for submitting your manuscript to PLOS ONE. After careful consideration, we feel that it has merit but does not fully meet PLOS ONE’s publication criteria as it currently stands. Therefore, we invite you to submit a revised version of the manuscript that addresses the points raised during the review process.

We look forward to receiving your revised manuscript.

Kind regards,

Sohaib Mustafa, Ph.D.

Academic Editor

PLOS ONE

Journal Requirements:

Reviewers' comments:

Reviewer's Responses to Questions

**Comments to the Author**

1. If the authors have adequately addressed your comments raised in a previous round of review and you feel that this manuscript is now acceptable for publication, you may indicate that here to bypass the “Comments to the Author” section, enter your conflict of interest statement in the “Confidential to Editor” section, and submit your "Accept" recommendation.

Reviewer #2: All comments have been addressed

Reviewer #3: (No Response)

2. Is the manuscript technically sound, and do the data support the conclusions?

Reviewer #2: Yes

Reviewer #3: Yes

3. Has the statistical analysis been performed appropriately and rigorously? 

Reviewer #2: Yes

Reviewer #3: Yes

4. Have the authors made all data underlying the findings in their manuscript fully available?

Reviewer #2: Yes

Reviewer #3: Yes

5. Is the manuscript presented in an intelligible fashion and written in standard English?

Reviewer #2: Yes

Reviewer #3: Yes

6. Review Comments to the Author

Reviewer #2: (No Response)

Reviewer #3: Please find my remarks below:

1) Please highlight the novelty of the work.

2) What are the limitations of the work?

7. PLOS authors have the option to publish the peer review history of their article (what does this mean?). If published, this will include your full peer review and any attached files.

Reviewer #2: **Yes: **Anurag Tiwari

Reviewer #3: No

---

## [Author Response · Author response to Decision Letter 2]

7 Jan 2025

Resubmission 6th January 2025: As requested, we have highlighted the novelties and limitations of the paper. These changes have been marked in the version with track changes and also emphasized in the rebuttal letter.

---

## [Decision Letter · Decision Letter 2]

23 Jan 2025

Algorithm, Expert, or Both? Evaluating the Role of Feature Selection Methods on User Preferences and Reliance

PONE-D-24-30693R2

Dear Dr. Kornowicz,

We’re pleased to inform you that your manuscript has been judged scientifically suitable for publication and will be formally accepted for publication once it meets all outstanding technical requirements.

Kind regards,

Sohaib Mustafa, Ph.D.

Academic Editor

PLOS ONE

Additional Editor Comments (optional):

Reviewers' comments:

Reviewer's Responses to Questions

**Comments to the Author**

1. If the authors have adequately addressed your comments raised in a previous round of review and you feel that this manuscript is now acceptable for publication, you may indicate that here to bypass the “Comments to the Author” section, enter your conflict of interest statement in the “Confidential to Editor” section, and submit your "Accept" recommendation.

Reviewer #3: All comments have been addressed

2. Is the manuscript technically sound, and do the data support the conclusions?

Reviewer #3: Yes

3. Has the statistical analysis been performed appropriately and rigorously? 

Reviewer #3: N/A

4. Have the authors made all data underlying the findings in their manuscript fully available?

Reviewer #3: Yes

5. Is the manuscript presented in an intelligible fashion and written in standard English?

Reviewer #3: Yes

6. Review Comments to the Author

Reviewer #3: I am happy with the revisions made to the revised manuscript. The authors have adequately addressed all the queries I raised.

7. PLOS authors have the option to publish the peer review history of their article (what does this mean?). If published, this will include your full peer review and any attached files.

Reviewer #3: No

---

## [Editor Report · Acceptance letter]

PONE-D-24-30693R2

PLOS ONE

Dear Dr. Kornowicz,

I'm pleased to inform you that your manuscript has been deemed suitable for publication in PLOS ONE. Congratulations! Your manuscript is now being handed over to our production team.

Kind regards,

on behalf of

Dr. Sohaib Mustafa

Academic Editor

PLOS ONE